# Accounting for corner flow unifies the understanding of droplet formation in microfluidic channels

Piotr M. Korczyk [1,5], Volkert van Steijn [2,5], Slawomir Blonski [1], Damian Zaremba [1], David A. Beattie [3] & Piotr Garstecki [4]

While shear emulsification is a well understood industrial process, geometrical confinement in microfluidic systems introduces fascinating complexity, so far prohibiting complete understanding of droplet formation. The size of confined droplets is controlled by the ratio between shear and capillary forces when both are of the same order, in a regime known as jetting, while being surprisingly insensitive to this ratio when shear is orders of magnitude smaller than capillary forces, in a regime known as squeezing. Here, we reveal that further reduction of—already negligibly small—shear unexpectedly re-introduces the dependence of droplet size on shear/capillary-force ratio. For the first time we formally account for the flow around forming droplets, to predict and discover experimentally an additional regime—leaking. Our model predicts droplet size and characterizes the transitions from leaking into squeezing and from squeezing into jetting, unifying the description for confined droplet generation, and offering a practical guide for applications.

[1] Institute of Fundamental Technological Research, Polish Academy of Sciences, Pawinskiego 5B, 02-106 Warsaw, Poland. [2] Faculty of Applied Sciences, Department of Chemical Engineering, Delft University of Technology, van der Maasweg 9, 2629 HZ Delft, The Netherlands. [3] Future Industries Institute, University of South Australia, Mawson Lakes Campus, Mawson Lakes, SA 5095, Australia. [4] Institute of Physical Chemistry, Polish Academy of Sciences, Kasprzaka 44/52, 01-224 Warsaw, Poland. [5] These authors contributed equally: Piotr M. Korczyk, Volkert van Steijn. Correspondence and requests for materials should be addressed to P.M.K. (email: piotr.korczyk@ippt.pan.pl) or to V.v.S. (email: v.vansteijn@tudelft.nl) or to P.G. (email: garst@ichf.edu.pl)

In spite of the beautiful regularity in flows of droplets in microfluidic networks at low Reynolds and capillary numbers, their dynamics offers rich phenomenological complexity[1–4] that prohibits predictive understanding. A striking example is the flow of a single droplet through a microchannel[5–8] for which the most basic question how the speed of the droplet depends on flow conditions, fluid properties, and the level of confinement still lacks a full answer. The generation of droplets in micro-channels[9,10] using T-junctions[11–19], flow-focusing[20–22], co-flow[23], step emulsification[24–29] and parallel devices[30–32], opened the new discipline of droplet microfluidics[33,34] that revolutionized analytical methods in biology[35] and medicine with digital assays[36], single cell sequencing[37], or systems for research on biological evolution[38]. Droplet microfluidic systems are also used to create new materials for pharmaceutical[39–41], cosmetics[42] and food[43] industries. In a stark contrast to the bulk process of shear emulsification that is one of the more illustrative and simple textbook examples of dimensional analysis, generation of droplets in confinement is still not completely understood.

The first microfluidic device used for the generation of dro-plets, a T-junction, was proposed by Thorsen et al.[11], who demonstrated that the dynamics of droplet formation is generally governed by surface tension and viscous shear, while body forces such as inertia or gravity play little role[9,13]. Depending on the relative magnitude of surface tension and shear, as captured by the capillary number (Ca), and on the contrast of viscosities between the two phases, distinct "visco-capillary" regimes have been identified[10]: dripping[13,14,16,22,44,45], jetting[44–46] and parallel co-flow[47]. Soon after, it was discovered that "capillary-domi-nated" formation of droplets in microconfinement results in droplet sizes that only depend of the ratio of the flow rates of the two immiscible liquids, completely independent of Ca, as described by the squeezing model[12,48]. This simple relation between droplet size and flow rates as well as the low poly-dispersity of the generated droplets makes the squeezing regime attractive for applications, where high precision and reproduci-bility are required in combination with independence on material parameters such as, e.g., the viscosity of the sample liquid. Effi-cient use of this technique thus requires a good understanding of the limits of the squeezing regime. However, the squeezing model, while commonly accepted, does not account for the flow of continuous liquid past the droplet while it is formed.

We show in this paper that the neglect of this corner flow entails spectacular failure of the squeezing model for vanishing values of capillary numbers. By formally accounting for this leaking flow we predict and verify experimentally a number of new features of generation of droplets in microconfinement, including an additional leaking regime at the lowest values of capillary number, the existence of a previously unknown lower bound of the squeezing regime, and scaling of the upper bound— the transition from squeezing to jetting. The model that we here demonstrate offers the unique attempt to a unified mechanistic description of the dynamics of droplet formation in microfluidic confinement.

## Results

**Experimental evidence for the leaking regime.** We studied the formation of droplets in the commonly used geometry of the-so-called T-junction[11] (Fig. 1a). Our device comprises a perpendi-cular intersection of two inlet channels that deliver two immis-cible liquids, the droplet phase (DP) and the continuous phase (CP), and a common output from the junction. The CP pre-ferentially wets the walls of the channels and the droplets never contact the walls, always being separated from them by at least a thin film of the CP. This prevents pinning of a contact line, which

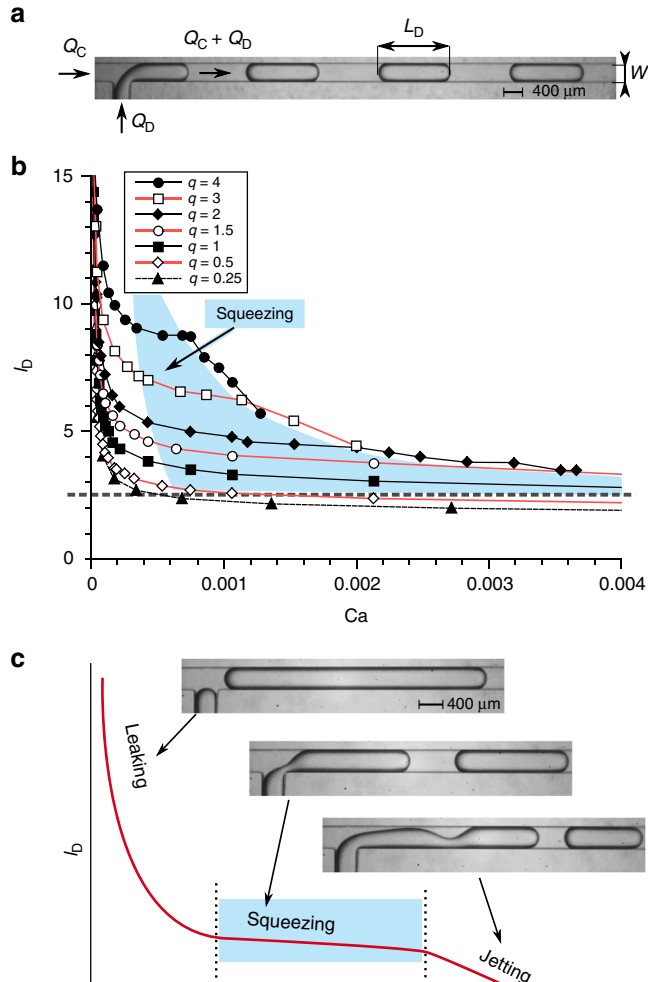

**Fig. 1** Droplet formation at a microfluidic T-junction and experimental data for the length of droplets versus the capillary number. **a** Snapshot illustrating the geometry of the T-junction with channels of a square cross section, i.e. $W = H = 360\,\mu m$. $Q_C$ and $Q_D$ are the flow rates of the CP (hexadecane) and DP (fluorinated oil FC-40), respectively. $L_D$ is the length of a droplet. **b** Experimental data—normalized length of a droplet $l_D = L_D/W$ as a function of the capillary number Ca for different $q = Q_D/Q_C$. The blue area highlights the region, where all curves have a plateau, interpreted as the squeezing regime. The boundary at the bottom of this area is taken as the minimum length for which squeezing is operative ($l_D \approx 2.5$, dashed line). **c** Schematic picture of the scaling of the normalized length of a droplet, $l_D$, with the capillary number Ca, as extracted from the full experimental data set in **b**, illustrating the leaking, squeezing, and jetting regimes

would render the dynamics irreproducible and hard to control. Stationary inflow of both phases into the junction causes a per-iodic breakup of the DP into droplets.

We measured the length of droplets as a function of Ca, for various ratios $q = Q_D/Q_C$ of flow rates of the DP and CP (see Fig. 1b). We normalized the length of the droplets, $L_D$, using the width of the channel, $W$, as $l_D = L_D/W$, and defined the capillary number as $Ca = \mu_C U/\gamma$, with $\mu_C$ the dynamic viscosity of the CP, $U = Q_C/HW$ the mean speed of the CP, $H$ the channel height, and $\gamma$ the liquid–liquid interfacial tension. The qualitative behaviour obtained from these measurements is illustrated in Fig. 1c. The plateau in which droplet length is virtually independent of Ca,

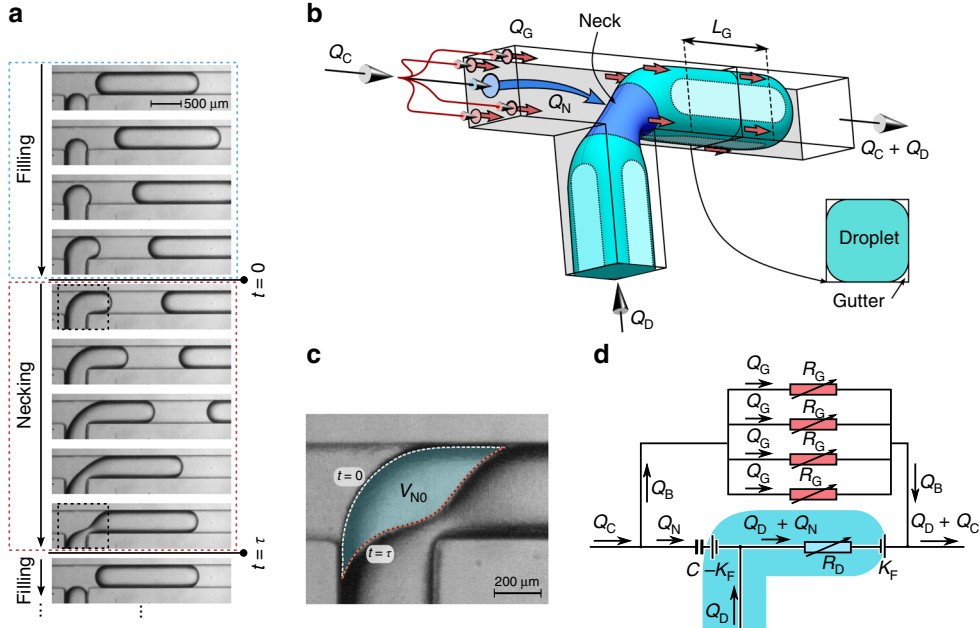

**Fig. 2** The process of droplet formation and concepts behind the theoretical model. **a** Consecutive snapshots showing a complete droplet formation cycle comprising a filling and a necking stage. **b** 3D schematic view of the geometry of a forming droplet in the necking stage showing the decomposition of the incoming CP flow ($Q_C$) in a flow towards the neck ($Q_N$) and four flows through the gutters ($Q_G$). **c** Comparison of shapes of the neck at the start of the necking stage ($t = 0$) and just before the neck breaks ($t = \tau$), (both images were extracted from the highlighted rectangles in the snapshots in **a**). The difference of these shapes defines the volume of the neck $V_{N0}$, which must be filled by the continuous phase to induce pinch-off. **d** Circuit diagram illustrating how flows towards and around the forming droplet depend on the time-dependent resistances ($R_G$ for viscous resistance in a gutter and $R_D$ for viscous resistance inside the forming droplet). Laplace pressure jump at the front of a droplet is shown schematically as an 'electromotive force' established by the curvature of the interface $K_F$. The change of the Laplace pressure jump due to the accumulation of the CP behind the forming droplet is modelled as the combination of a capacitance $C$ and the electromotive force with reversed direction in respect to the front of a droplet $-K_F$

confirms the well-accepted squeezing regime[12]. The Ca-dependent region at higher Ca indicates the—also known—jetting regime. The most intriguing aspect of the data in Fig. 1b, c is in the range of vanishing Ca, where the length of droplets explodes with Ca → 0. The squeezing regime not only spans a narrower range of Ca than previously expected[13] due to the existence of the lower boundary, but also due to the dependence of the upper boundary on $q$. Generally, the higher $q$, the narrower the range of Ca for the squeezing regime, as clearly illustrated by the highlighted area in Fig. 1b. Given the strong Ca-dependence in the here identified leaking regime, a good understanding of the mechanism that introduces this dependency is crucial for practical applications and presented next.

**Mathematical model of the leaking regime**. The starting point for our theoretical framework is the original squeezing model[12,48], which considers droplet formation as a two-step process (Fig. 2a). During the first 'filling' stage, that starts when the previous droplet has detached, the tip of the DP expands into the main channel and fills most of the junction. In the second, 'necking' stage, the droplet grows, extending downstream from the junction, while the CP squeezes the 'neck' (Fig. 2b). Defining the volume occupied by the neck, $V_N(t)$, with respect to the shape of the neck at pinch-off, this volume gradually decreases from $V_{N0}$ at the start of the necking stage ($t = 0$) to zero at pinch-off ($t = \tau$) (see Fig. 2c). The space left behind the moving interface is filled by the incoming volume of CP: $V_N^*(t) = V_{N0} - V_N(t)$. The final volume of a droplet, $V_D$, can be decomposed as the volume at the end of the filling stage, $V_{fill}$, and the volume added at a rate $Q_D$ during the time $\tau$ of the necking stage. Hence, $V_D = V_{fill} + Q_D\tau$. The original squeezing model assumes complete blockage of the channel by the forming droplet during the necking stage and calculates the necking time

as—simply—the time required for the continuous phase to displace the volume initially occupied by the neck $V_{N0}$, i.e. $\tau = V_{N0}/Q_C$ (see Fig. 2c). This assumption overlooks that a non-wetting droplet does not fill the corners of a channel that has a rectangular cross section[49,50], allowing the CP to flow (leak) by the droplet through these corners, the so-called 'gutters'[15] (see Fig. 2b). For an elongated droplet steadily pushed through a straight rectangular channel, Wong et al.[5] pointed out that such a droplet acts as a leaky piston with the fraction of the incoming CP flowing around the droplet (through the gutters) increasing as Ca → 0. At low Ca, it is hence expected that the fraction of incoming liquid that passes by a forming droplet, and thus does not contribute to the squeezing, is no longer negligible. This introduces a Ca-dependence in the duration of the necking stage and hence qualitatively explains the here observed Ca-dependence of the volume of the droplets at Ca → 0. In channels without gutters, this Ca-dependence hence should be absent. Indeed, additional experiments using a T-junction with 'gutter-free' circular channels reveal that the length of the droplets varies weakly with Ca in comparison to T-junctions with square channels (see Supplementary Note 1 and Supplementary Fig. 1). An interesting complication in the description of corner (or gutter) flow around a forming droplet—as compared with a droplet moving steadily through a straight channel—is that the gutter flow is dynamic due to the simultaneous change in its driving force (interface curvature) and in its resistance to flow (length of gutters). Although earlier work did assume a fixed, non-zero, fraction of the CP stream to flow around a forming droplet[15], here we introduce the functional dependence of the leaky flow through the gutters on fluid properties and flow conditions to establish a unifying description.

Specifically, we define the instantaneous flow rate of the CP through a single gutter as $Q_G(t)$ and through all four

gutters combined as $Q_B(t) = 4Q_G(t)$ (see Fig. 2b, d). Then, the flow contributing to squeezing of the neck is $Q_N(t) = Q_C - Q_B(t) = dV_N^*/dt = -dV_N/dt$. The necking time thus equals $\tau = V_{N0}/\bar{Q}_N$, with $\bar{Q}_N$ being the time-averaged squeezing rate. Introducing $\eta = Q_B/Q_N$ as the relative leaking strength and $\bar{\eta}$ as its time average, we obtain $\tau = \frac{V_{N0}}{Q_C}(1 + \bar{\eta})$. The droplet volume hence becomes $V_D = V_{fill} + qV_{N0}(1 + \bar{\eta})$. Rewritten in terms of the non-dimensional length $l_D = L_D/W$, using $L_D \approx V_D/HW$, as is valid for long droplets[51], we obtain:

$$l_D = l_0 + qv_{N0}(1 + \bar{\eta}) \tag{1}$$

with $l_0 = V_{fill}/HW^2$ and $v_{N0} = V_{N0}/HW^2$. This analysis generalizes the original squeezing model, recovered for $\bar{\eta} = 0$. In order to quantitatively predict the size of the droplets in the leaking regime, we next derive the functional dependence of $\bar{\eta}$, on $q$ and on Ca.

We start from the flow scheme depicted in Fig. 2d and note that the pressure difference associated with viscous flow of the CP through the gutters balances the pressure difference arising from the sum of viscous flow inside the DP and the difference in curvature of the interface at the front and at the back of the forming droplet. For the leaking and squeezing regimes (i.e. for low Ca), viscous shear is unable to deform the interfaces such that the Laplace law is used to calculate the pressure difference due to a difference in curvature ($K_F$ versus $K_B$) of the quasi-static interfaces as $\gamma(K_F - K_B)$. The viscous pressure head over the droplet and over the gutters equals $R_D(Q_D + Q_N)$ and $R_G Q_B/4$, respectively, with $R_D$ and $R_G$ the hydrodynamic resistances of the droplet and the gutter. The balance hence equals $\gamma(K_F - K_B) + R_D(Q_D + Q_N) = R_G Q_B/4$. For systems with a moderate viscosity contrast, $\mu_D/\mu_C$, the viscous pressure difference inside the droplet can be neglected with respect to that in the gutters, because the cross-sectional area of the gutters $A_G$ is much smaller compared with that of the droplet $A_D$. We hence continue with the simplified balance:

$$\gamma(K_F - K_B) = R_G Q_B/4. \tag{2}$$

Considering the right-hand side of Eq. (2), the hydrodynamic resistance of a gutter ($R_G$) increases proportionally to the length $L_G$ of the gutter: $R_G = \frac{\alpha_G \mu_C}{A_G^2} L_G$, with $A_G$ the cross-sectional area of a gutter (in good approximation constant along the gutter and independent of time) and $\alpha_G$ a dimensionless geometrical factor[52]. Referring to Fig. 2d, we estimate the velocity of the front of the droplet as $(Q_D + Q_N)/A_D$ and that of the back of the droplet as $Q_N/A_D$, such that the length of the gutter increases at a rate equal to the velocity difference, i.e. $\frac{dL_G}{dt} = Q_D/WH$, where we approximate the cross-sectional area of the droplet ($A_D$) with the area of the lumen of the channel $WH$. Considering the left-hand side of Eq. (2), we note that the curvature at the front of the droplet, $K_F = \frac{2}{W} + \frac{2}{H}$, is approximately constant in time. The curvature at the rear (i.e., at the neck) depends on the neck shape, which is determined by the amount of the CP collected behind it, i.e. by the value of the volume $V_N^*$. Adopting the electric–hydraulic analogy that can be applied to single phase low Reynolds number flows, we describe the accumulation of the CP behind the droplet akin to a charging capacitor: the further the rear interface is pushed into the junction, the larger the difference in curvatures at the rear and front, and hence the more charge is stored. If one were able to instantaneously release the driving pressures in the system during the formation of a droplet (akin to switching off the main voltage supply in an electric circuit), the capacitor would discharge, i.e., the forming droplet would relax its shape inside the T-junction to an equilibrium shape with similar curvatures at its front and rear.

Similar to electric RC circuits, the product of the capacitance and resistance can be seen as the characteristic relaxation time. A quantitative analysis of this time scale is provided in Supplementary Note 2. Using this analogy, we model the curvature difference as: $K_F - K_B = \frac{V_N^*}{C} = \frac{1}{W^3 H}\frac{V_N^*}{c} = \frac{1}{W}\frac{v_N^*}{c}$, with $C$ a constant (analogous to capacitance) characterising the rate of change of the curvature with $V_N^*$. Here $c = \frac{C}{W^3 H}$ and $v_N^* = \frac{V_N^*}{W^3 H}$ are dimensionless equivalents of $C$ and $V_N^*$, respectively. Before using the thus obtained relation, we provide an intuitive physical interpretation of the capacitance by connecting it to the classical derivation of the Young-Laplace law. Considering a droplet of volume, $V$, and surface area $A$, the Helmholtz free energy, $dF$, equals $dF = -pdV + \gamma dA$. Near equilibrium ($dF = 0$), the pressure difference across the droplet interface equals $\Delta p = \gamma \frac{dA}{dV} = \gamma K$. The curvature difference for a forming droplet, $K_F - K_B$, being equal to $\frac{V_N^*}{C}$ can similarly be related to $\frac{dA_N(V_N^*)}{dV_N^*}$, i.e. the change in neck area, $A_N$, parameterized solely in terms of $V_N^*$. We hence obtain $\frac{dA_N(V_N^*)}{dV_N^*} \sim \frac{V_N^*}{C}$, with the corresponding interfacial free energy, $\mathcal{W}$, being equal to $\mathcal{W} = \gamma A_N(V_N^*) = \frac{1}{2}\frac{(V_N^*)^2}{C}$. The capacitance $C$ can hence be interpreted as the coefficient that describes how the surface area of the neck, $A_N$, and the interfacial free energy stored in it due to its deformation by the CP, depends on the volume of the CP collected behind the neck. Now, we proceed with the main analysis by substituting the relation $K_F - K_B = \frac{1}{W}\frac{v_N^*}{c}$ in the pressure balance (Eq. (2)) together with $Q_B = Q_C - Q_N = Q_C - dV_N^*/dt$, to obtain: $\frac{\gamma V_N^*}{W^3 Hc} = R_G(t) \cdot (Q_C - dV_N^*/dt)/4$. This equation resembles the well-known equation for electric RC systems ($\frac{Q}{C} = -R \cdot dQ/dt$, with $Q$ the electric charge), although with a time-dependent resistance $R_G(t)$. The solution gives the flow rate to the neck: $Q_N = \frac{qCa}{qCa + \beta}Q_C$, with constant $\beta = \frac{4}{\alpha_G}\frac{A_G^2}{W^3 H}\frac{1}{c}$. Note that the full expression for $Q_N$ includes an additional time-dependent term that decays fast and can be neglected (see Supplementary Note 2). Strikingly, this model predicts that $Q_N$ does not depend on time, implying that, after a short initial transient, the flow through the gutters is constant in time. Consequently, the front of a forming droplet should propagate at a fixed speed during droplet formation[22]. Our experiments confirm this surprising prediction (see Supplementary Note 3 and Supplementary Fig. 2). Finally, we arrive at the following expression for the relative leaking strength: $\bar{\eta} = \eta = \frac{Q_B}{Q_N} = \frac{Q_C - Q_N}{Q_N} = \frac{\beta}{qCa}$. Incorporating this functional dependence in Eq. (1), we obtain:

$$l_D = l_0 + qv_{N0}\left(1 + \frac{\beta}{qCa}\right) \tag{3}$$

This generalized equation implies that all data from Fig. 1b should collapse onto a single master curve when using this functional form for the dependence of droplet size on the ratio of flow rates and the capillary number. Indeed, fitting a single set of parameters $\beta$, $v_{N0}$, and $l_0$ to all our data, we find a single master curve for $\bar{\eta} = \frac{\beta}{qCa} = \frac{l_D - l_0}{qv_{N0}} - 1$ as shown in Fig. 3. We confirm the universality of this behaviour for different fluid systems and channel sizes (see Supplementary Notes 4, 5 and Supplementary Figs. 3, 4). For rectangular channels with different aspect ratios, we confirm that the general behaviour is the same, with the details of the leaking mechanisms depending on the aspect ratio (Supplementary Note 6, Supplementary Fig. 5 and Supplementary Table 4). With the functional behaviour of the leaking strength well captured, the generalized Eq. (3) accurately describes the experimental data on droplet length (see also Supplementary

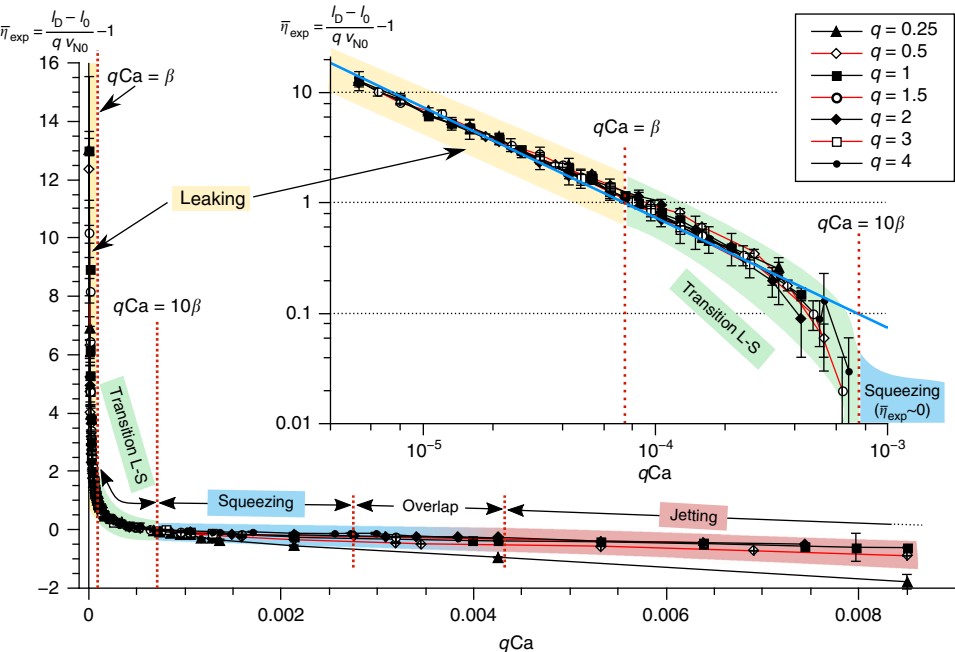

**Fig. 3** Experimental verification of the leaking-regime-model. Master curve for $\bar{\eta}$ experimentally determined from all data in Fig. 1b using $\overline{\eta_{\exp}} = \frac{l_D - l_0}{q\,v_{N0}} - 1$ (Eq. 3) and $\overline{\eta_{\exp}} = \frac{\beta}{q\mathrm{Ca}}$ (definition) with fit parameters $l_0 = 1.46 \pm 0.14$, $v_{N0} = 2.04 \pm 0.11$, and $\beta = 7.4 \times 10^{-5} \pm 0.3 \times 10^{-5}$. In the leaking regime, the flow through the gutters is at least equal to the flow to the neck (i.e. $\overline{\eta_{\exp}} \geq 1$ for $q\mathrm{Ca} \leq \beta$). By contrast, in the squeezing regime, the flow though the gutters is negligibly small compared with the flow to the neck (i.e. $\overline{\eta_{\exp}} \ll 1$ for $q\mathrm{Ca} > 10\beta$). The intermediate regime ($\beta < q\mathrm{Ca} < 10\beta$) is identified as the transition between leaking and squeezing. Inset: log-log master curve for $q\mathrm{Ca} < 10\beta$. Blue solid line: theoretical scaling $\bar{\eta} = \frac{\beta}{q\mathrm{Ca}}$. All experimental data collapse on this single curve in the leaking regime, while deviations of $\overline{\eta_{\exp}}$ from the theoretical scaling for $\bar{\eta}$ at $\overline{\eta_{\exp}} < 0.1$ are attributed to a remaining dependency of $v_{N0}$ (taken constant here) on $q$ and Ca as explained later. The transition between squeezing and jetting does not occur at a single value of $q\mathrm{Ca}$ for curves with different $q$, which is highlighted in the graph in the 'overlap' region. In the next part, we will explain how to parameterize the system to capture the squeezing–jetting transition for all these curves based on a single parameter

Note 7 and Supplementary Fig. 6). This agreement demonstrates that the present analysis captures the mechanisms governing droplet formation in the leaking regime. Figure 3 also shows that the leaking regime transitions smoothly into the squeezing regime. Without a sharp boundary, the lower limit of the squeezing regime, in which droplet length does not significantly depend on Ca, may be defined as $q\mathrm{Ca} \gg \beta$, as evident from Eq. (3). In case a particular application requires the sensitivity of the final droplet length (with respect to Ca) to be less than a threshold value of say 10% ($\bar{\eta} < 0.1$), $q\mathrm{Ca} = 10\beta$ provides an application-tailored lower limit for the squeezing regime.

**Scaling of the squeezing to jetting transition**. To explain why $\overline{\eta_{\exp}} = \frac{l_D - l_0}{q\,v_{N0}} - 1$ deviates from the theoretical scaling $\bar{\eta} = \frac{\beta}{q\mathrm{Ca}}$ at the largest values of $q\mathrm{Ca}$, we take a closer look at the shapes of the rear of the droplet at end of the necking stage. For the capillary-dominated regime, we expect self-similar shapes of the interface—that is, shapes that solely depend on the instantaneous volume of the neck $V_N^*(t)$. Considering the shape of the interface at the moment of pinch-off, we observe that the shapes are indeed all similar at low $q\mathrm{Ca}$ (Fig. 4a). The volume filled by the CP during the necking stage, $V_{N0}$—being directly related to this shape—is the same for all these cases, and so is $\beta$. In contrast, the interface shape prior to pinch-off is no longer self-similar at higher $q\mathrm{Ca}$, i.e. no longer determined solely by $V_N^*$. Viscous deformation of the interface, so far left out of the description, hence introduces a dependence on $q$ and on Ca in $v_{N0}$, explaining the deviation from Eq. (3) at high $q\mathrm{Ca}$, as observed in the inset of Fig. 3.

We hypothesize that, for the larger values of $q\mathrm{Ca}$, the instantaneous neck shape is not solely parameterized by $V_N^*$, as

in the case for low $q\mathrm{Ca}$, for which the final shape at pinch-off (for $V_N^*(t = \tau)$) has a universal value (though with different necking times $\tau$ for different conditions). To capture how viscous dissipation alters $V_N^*(t = \tau)$ beyond the leaking regime, we therefore introduce an additional parameter, which we call the 'shape number' $S$.

We derive an expression for $S$ using Onsager's variational principle[53,54], which is an extension of Rayleigh's least energy dissipation principle[55]. Onsager's principle allows for taking into account the dynamic change in free energy of the interface as well as the energy dissipated by viscous flow. It has been successfully applied in the field of soft matter physics[56]. In isothermal systems, Onsager's principle minimizes the so-called Rayleighian $\mathcal{R}$, which takes the following general form[57]:

$$\mathcal{R} = \Phi(x, \dot{x}) + \dot{\mathcal{W}}(x, \dot{x}). \tag{4}$$

Here, $\mathcal{W}$ is the interfacial free energy of the system with $\dot{\mathcal{W}}$ being its rate of change. $\Phi$ is the dissipation function, which is equal to half the rate of energy dissipation. $x$ represents the set of variables of the system and $\dot{x}$ their time derivatives. In our system, viscous dissipation is the only source of energy dissipation. The rate of viscous dissipation for laminar flow can generally be expressed in terms of the hydrodynamic resistance ($R$) and the flow rate ($Q$) as $RQ^2$ [58]. In our system, for the larger values of $q\mathrm{Ca}$, viscous dissipation mainly stems from the flow through the neck. For that $q\mathrm{Ca}$ range, gutter flows are negligibly small and the neck is squeezed by almost all of the incoming CP ($Q_N \approx Q_C$). The flow rate in the neck itself hence equals the sum $Q_C + Q_D$. The viscous resistance in the neck, $R_N$, depends on its instantaneous shape and hence is parameterized by $V_N^*$ and $S$. The dissipation

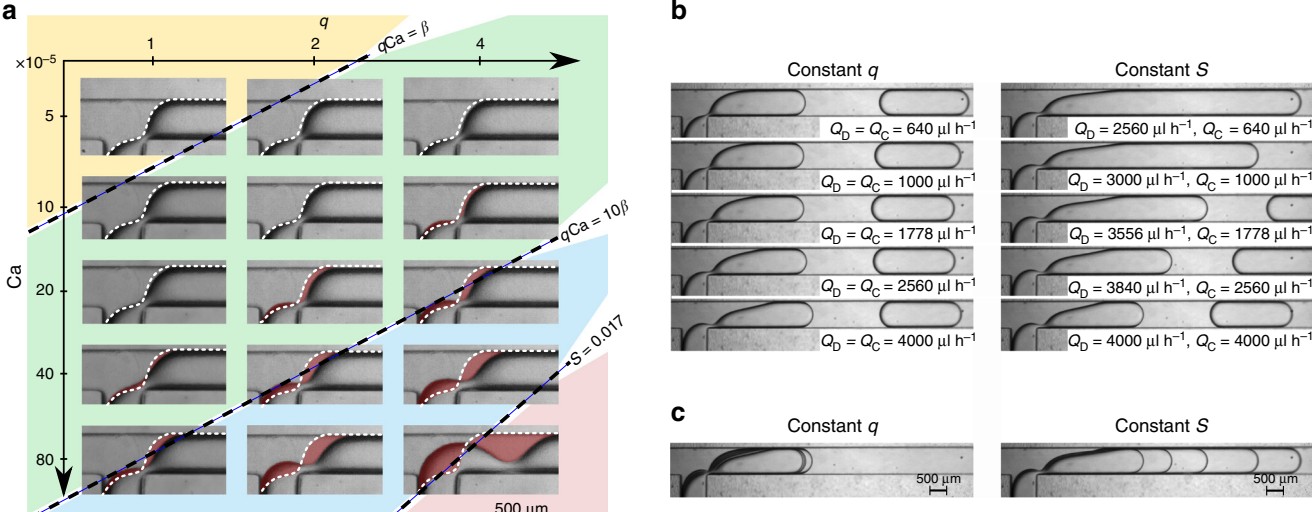

**Fig. 4** Variations of the shape of the neck. **a** The neck just before breakup for different values of Ca and $q$ in a T-junction with channels of a square cross section ($W = H = 360$ µm). All shapes are compared with the shape in the top left corner (white dashed line) with the differences highlighted in colour. The diagonal lines separating the snapshots correspond to $q$Ca $= \beta$ and $q$Ca $= 10\beta$. For the leaking regime ($q$Ca $< \beta$; yellow) and for the transition between leaking and squeezing ($q$Ca $< 10\beta$; green), this clearly shows that the shape prior to pinch-off is the same irrespective of the values of $q$ and Ca, confirming that the parameters depending on the shape such as $v_{NO}$, $K_B(V_{NO}^*)$ and $K_B(0)$ are constant in the leaking regime and in the transition regime between leaking and squeezing. For the squeezing regime ($q$Ca $> 10\beta$; blue) and the jetting regime ($S > 0.017$; red), the remaining dependency of the shape on $q$ and Ca is clearly visible. **b** Comparison of shapes of the neck for a variety of combinations of $Q_C$ and $Q_D$. For better visualisation, a T-junction with flattened channels ($W = 2H = 800$ µm) was used. The left column—constant $q(=1)$, the right one—constant $S(=0.017)$. Snapshots in the same row were taken for the same $Q_C$ (i.e. the same Ca). **c** Composition of overlaid images from each column from **b**. It is visible that, although the length of a droplet is similar for constant $q$ (a well-known feature of the squeezing regime), the shapes of the neck differ significantly. By contrast, these shapes are the same for constant $S$

function hence is estimated as $\Phi = \frac{1}{2}R_N(V_N^*, S) \cdot (Q_C + Q_D)^2$. The free energy of the system $\mathcal{W}$ is equal to the interfacial energy of the neck $\mathcal{W}(V_N^*, S) = \gamma A_N(V_N^*, S)$, where $A_N(V_N^*, S)$ is the instantaneous surface area of the neck. The rate of change of the free energy can be estimated as $\dot{\mathcal{W}}(V_N^*, S) = \gamma \frac{\partial A_N(V_N^*, S)}{\partial V_N^*} \frac{\partial V_N^*}{\partial t} = \gamma K(V_N^*, S)Q_N \approx \gamma K(V_N^*, S)Q_C$, with $K(V_N^*, S)$ the instantaneous curvature of the neck. We hence obtain the following Rayleighian

$$\mathcal{R} = \frac{1}{2}R_N(V_N^*, S)(Q_C + Q_D)^2 + \gamma K(V_N^*, S)Q_C \quad (5)$$

Solving the minimization equation for the Rayleighian with respect to $S$ for fixed $V_N^*$, $\frac{d\mathcal{R}}{dS}\big|_{V_N^*} = 0$, we obtain:

$$2\frac{k_S(V_N^*)}{r_S(V_N^*)} = \frac{(Q_C + Q_D)^2}{Q_C} \propto \text{Ca}(1 + q)^2 \quad (6)$$

where $k_S(V_N^*) = \gamma \frac{dK(S, V_N^*)}{dS}\big|_{V_N^*}$ and $r_S(V_N^*) = -\frac{dR_N(S, V_N^*)}{dS}\big|_{V_N^*}$, with $k_S(V_N^*)$ and $r_S(V_N^*)$ both defined as positive functions based on the expectation that larger $S$ implies larger deformation of the interface (and hence larger $K(S, V_N^*)$) and smaller viscous resistance (and hence smaller $R_N(S, V_N^*)$). With the left-hand side of Eq. (6) only depending on the shape of the neck, this analysis teaches that the neck shape is fully governed by $V_N^*$ and the parameter Ca$(1 + q)^2$, which we call the 'shape number' defined as $S = \text{Ca}(1 + q)^2$. We hence expect the same neck shape prior to pinch-off ($V_N^*(t = \tau)$) for different conditions, as long as $S$ is the same. Before we test the validity using experiments, we stress that the same analysis can be applied for low $q$Ca with the neck shape solely parameterized in terms of $V_N^*$, with the Rayleighian being equal to $\mathcal{R} = \frac{1}{2}R_G \cdot \frac{Q_B^2}{4} + \frac{V_N^*}{C}(Q_C - Q_B)$. Minimizing $\mathcal{R}$ with respect to $Q_B$ for fixed $V_N^*$, we obtain the same balance $\left(\frac{V_N^*}{C} = R_G \cdot \frac{Q_B}{4}\right)$ as the one

derived using the momentum balance as a starting point, confirming the validity of the approach.

The above analysis predicts that, for the larger range of $q$Ca, all neck shapes prior to pinch-off are uniquely defined by the value of $S$. Remarkably, this is exactly what we find in experiments, with all interface shapes collapsing onto a master shape for fixed $S$, as evident from Fig. 4b, c. Having established the physical origin of this $S$-number, we finalize the description of droplet formation. We use the $S$-number to predict the upper boundary for squeezing regime, and to explain the mechanism behind the transition between squeezing and jetting.

At the limit of high rates of flow in the squeezing regime, we expect shear to wash the instability away from the junction, as observed in the convective regime in jetting. Since this mechanism is not captured in the $S$-number description, we expect that, in that limit, the shape of the neck just before breakup (for $V_N^*(t = \tau)$) will no longer be uniquely captured by $S$. To test this hypothesis experimentally, we determined the relation between $V_N^*(t = \tau)$ and $S$. We use the breakup distance $D$ (see inset Fig. 5a) as a proxy for $V_N^*(t = \tau)$ as it is a well-defined direct observable. Data for different values of $q$ collapse perfectly on the same curve for $d = D/W \lesssim 1$ and $S \lesssim 0.017$. For larger $S$, the breakup shape is indeed no longer uniquely defined by $S$ as evident from the curves for different $q$. We therefore interpret this point ($S_{\text{crit}} \approx 0.017$) as the transition from squeezing to jetting, supported by the experimental observation that for larger $S$, a long thread of the DP penetrates the main channel, characteristic for the jetting regime. This threshold value of $S$ accurately predicts the transition for all curves (Fig. 5b). We note that, to the best of our knowledge, the $S$-number criterion is the first attempt to provide a universal scaling of the squeezing–jetting transition.

In summary, we derived a complete model for the generation of droplets at low Ca under microscale geometrical confinement. The explicit inclusion of the magnitude of the leaking flow of the continuous phase past a growing droplet allowed us

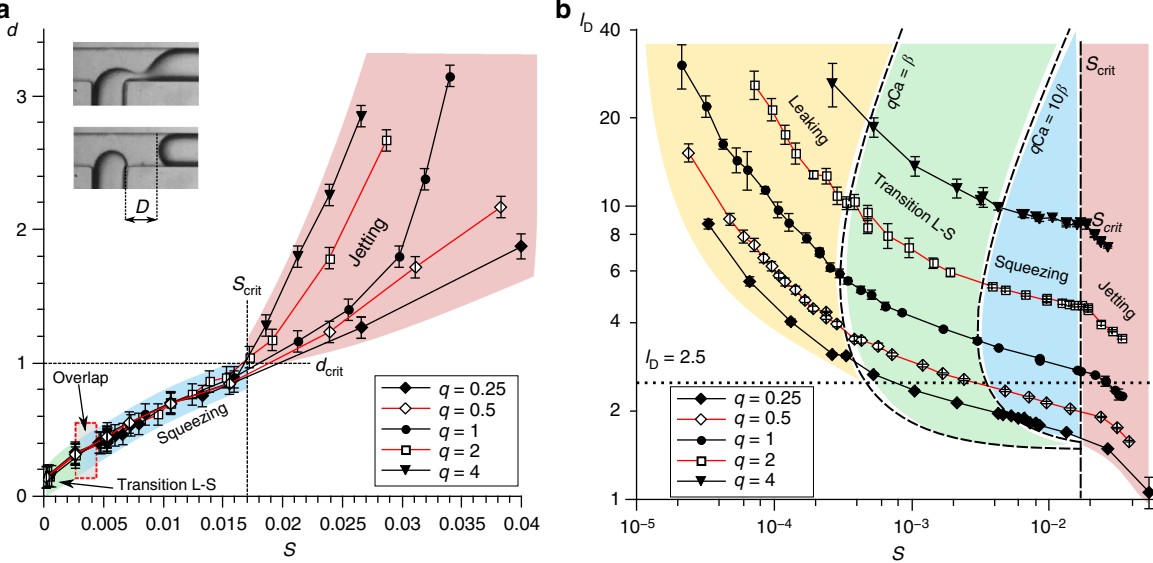

**Fig. 5** Quantitative test of the S-number as the scaling factor of the formation of droplets in the squeezing regime. **a** Measurements of the dimensionless distance $d = D/W$—the distance between the internal corner of the junction and the rear interface of the droplet immediately after breakup—as a function of the S-number. Inset: shape of the neck prior to breakup (top) and immediately after breakup (bottom). The measured value of $d$ corresponds to the deformation of the shape of the neck. The overlap between transition L–S and squeezing corresponds to the fact that the transition between these regimes is captured by a single value $q$Ca for all curves, but not for a single value of the S-number. **b** Normalized length of droplets $l_D$ versus the S-number showing that the transition from squeezing to jetting occurs at a single value for the S-number for all curves ($S_{crit} \sim 0.017$). For the series with $q < 1$ the length of the droplets (except for the leaking regime) is below or close to $l_D = 2.5$. In the case of such short droplets, a more accurate description should include shear during the filling stage as in models for the dripping regime

to uncover the leaking regime of drop formation and to reformulate the squeezing model for the size of the droplets, including the transitions between leaking and squeezing, and between squeezing and jetting. The insights from this work can be used to revisit droplet formation at low Ca in other common geometries, such as Y-junctions, cross-junctions, and flow-focusing devices, including many fine details such as the influence of channel aspect ratio and viscosity contrast that, to date, have escaped a unified model of droplet formation in microfluidic confinement.

From a practical point of view, the leaking regime is the least attractive mode of breakup, because it presents a very sharp dependence of droplet volume on the control parameters—such as rates of flow. One may either use the equations here described to find the squeezing regime for channels with a square cross section, or, easier, use a T-junction with circular channels that preclude leaking and present a very wide range of parameters that result in squeezing.

## Methods

**Device fabrication.** We fabricated the chips via direct milling in polycarbonate (PC) sheets (Macroclear, Bayer, Germany) using a CNC milling machine (Ergwind, Poland). This machine has a reproducibility of positioning of 5 μm. The milled chips were bonded to a flat slab of polycarbonate using a hot press at 130 °C. No further channel modifications were applied. The height and width of the rectangular channels was 360 μm, except for the chips used to produce Fig. 4b, c, Supplementary Fig. 5 ($W = 2H = 800$ μm) and Supplementary Fig. 4 ($W = H = 200$ μm). These dimensions, which may be larger than those typically encountered, were chosen to enable reaching low Ca values, without being constrained by the lower limit of the flow rate of the syringe pumps.

**Experiments.** We used a stereoscope equipped with a high-speed camera (PCO HS1200) to record images of droplet formation. We analyzed the sequences of images with a custom written script in MATLAB (Mathworks), which automatically recognized droplets and measured their length. In order to obtain data with a good precision, long sequences of droplets (typically > 30) were observed, which for the low Ca values took up to 10 h.

In order to feed our system with liquids, we used Nemesis pumps (Cetoni GmbH, Germany) with 100 μl glass syringes, connected to the chip using PE-60 tubing (Beckton-Dickinson, USA). In addition, we performed some measurements with 1000 μl syringes to test whether the pumps generate fluctuations in flow rates that could disturb the measurements[59]. The measurements from both syringes were in good agreement (see Supplementary Note 8 and Supplementary Fig. 7), confirming that there are no significant fluctuations in our feeding system. Having confirmed that the syringe pumps produce steady flows for our range of operating conditions, we have chosen their use over, for example, pressure driven systems, as they allow direct control over the flow rate.

We used fluorinated oil FC-40 (3M, USA) as DP and hexadecane (Sigma Aldrich Co.) as CP for the measurements reported in the main article. We chose this set after multiple attempts with different fluid combinations. The chosen fluid system ensured the absence of wetting of the channel walls by the DP, without further channel treatment or addition of surfactants. Dynamic effects resulting from surfactant transport[17] are hence not at play. Additional fluid systems used to construct Supplementary Fig. 3 are further detailed in Supplementary Note 4 and Supplementary Table 1.

Viscosities of the used liquids were estimated by measuring the time required for a given volume to flow through a calibrated capillary for a known pressure drop over the capillary, which was controlled using a pressure regulator and a precise manometer. We repeated the measurements for different values of the pressure drop obtaining a linear relation between pressure drop and calculated flow rate. A linear fit provided the values of viscosities. This resulted in $\mu_C = 3.6$ mPa s and $\mu_D = 4.1$ mPa s for hexadecane and FC-40, respectively. Values for the other fluid systems are reported in Supplementary Table 2.

The interfacial tension between both liquid phases was estimated by the pendant droplet method using a custom set-up enabling observation of the interface of a pendant droplet of FC-40 immersed into hexadecane. We calculated the interfacial tension by the use of a custom Matlab script applying the Laplace-Young theorem to the droplet shape extracted from the acquired images. This resulted in $\gamma = 7.3$ mN m$^{-1}$ for the hexadecane—FC-40 fluid system. Values for the other fluid systems are reported in Supplementary Table 3.

## Data availability

The data that support the plots within this paper and other findings of this study are available from the corresponding authors upon reasonable request.

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

## Acknowledgements

The project operated within the First Team grant (POIR.04.04.00-00-3FEF/17-00) of the Foundation for Polish Science co-financed by the EU under the Smart Growth

Operational Programme. V.vS. is supported by a Veni grant (13137) of NWO-STW. D.Z. acknowledges support within the grant Sonata-bis (2014/14/E/ST8/00578) of the National Science Centre, Poland. D.A.B. and P.M.K. acknowledge support from Marie Curie International Outgoing Fellowship within the 7th European Community Framework Programme (PIOF-GA-2011-302803). P.G. acknowledges support within the Foundation for Polish Science Team-Tech 2016-2/10 program. We thank Bartosz A. Grzybowski for critical reading of the manuscript and for helpful comments.

## Author contributions

P.M.K., V.vS. and P.G. designed the study. P.M.K., S.B. and D.Z. planned experiments and performed measurements. P.M.K., V.vS. and P.G. provided mathematical models. P.M.K. and V.vS. share the first authorship. P.M.K., V.vS., S.B., D.Z., D.A.B. and P.G. contributed to the preparation of the manuscript.

## Additional information

**Competing interests:** The authors declare no competing interests.

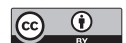 ns license, unless indicated otherwise in a credit line to the material. If material is not included in the article's Creative Commons license and your intended use is not permitted by statutory regulation or exceeds the permitted use, you will need to obtain permission directly from the copyright holder. To view a copy of this license, visit http://creativecommons.org/licenses/by/4.0/.

© The Author(s) 2019

