## [Peer Review File · Nature Communications]

Reviewers' comments:

Reviewer #1 (Remarks to the Author):

Title: "Accounting for corner flow unifies the understanding of droplet formation in microfluidic channels"

MS#: NCOMMS-18-35904-T

This study reports a new regime of droplet formation, leaking regime, which helps enrich the understanding of droplet formation at low Capillary number, the number that compares capillary force with viscous force. The finding of this new regime is made to account for the corner (gutter region of the channels of rectangular) flow and thus its shear impact on droplet formation. In general, the corner flow is neglected when modelling droplet formation in the squeezing regime (low capillary number). The authors found neglecting the corner flow results in failure in predicting droplet formation. This new findings is very impactful to the field and particularly impactful to the applications of droplet microfluidics which normally works with low Capillary numbers.

The study is systematically designed, precisely executed, and accurately and carefully analyzed. For example, when syringe pump is used for pumping the fluids, they carefully looked at its cyclic behavior on droplet formation by varying the syringe size significantly. Another example is that the findings of the leaking regime should disappear when no gutter region is present. The authors actually did perform such an experiment to confirm their findings, which is appreciated. Similar considerations for other aspects were made which is appreciated. The manuscript is well written. I would like to recommend it to be published with the following suggestions to be addressed.

General comments:

1. It's well known that syringe pumps have long-term and short-term oscillations in pumping and one of the authors actually reported the differences between syringe pumps and pressure systems for droplet formation, where pressure systems are much better in producing stable droplet size. It is clear and appreciated that the authors carefully checked the impact of the syringe pump used on droplet formation. The authors should comment on the reason of choosing syringe pumps versus pressure systems.
2. The authors should comment on the reason of choosing much larger channels (800 μm) for images shown in Figure 4 than most channels (360 μm) for other experiments. Was it for better images?
3. The channel dimensions for most experiments were 360 μm which seems larger than normally used in droplet microfluidics. The rational of choosing this dimension should be provided.
4. "very" in the sentence: "Indeed, additional experiments using a T-junction with 'gutter-free' circular channels reveal that the length of the droplets very weakly depends on Ca in comparison to T-junctions with square channels (see ESI, Fig. S.01)" should be "vary".

Carolyn Ren

Reviewer #2 (Remarks to the Author):

This paper reports the results of an experimental and scaling study of flows in microfluidic channels with T-junctions. These types of flows exhibit a number of regimes, such as "jetting" and "squeezing" amongst others. This work focuses on a certain regime referred to as "leaking", on deriving scalings that collapse their experimental data in terms of droplet size/length vs rescaled capillary numbers,

and provide transitions between the various regimes. These scaling results are possible by accounting for the fact that drops are not completely flush against the square channel walls; there are "gutter"-like regions surrounding the droplet through which a proportion of the continuous phase flow can bypass the droplet. This, therefore, implies that not all of the continuous phase liquid can contribute to the squeezing that gives rise to the pinch-off event, which, in turn, leads to the formation of a discrete drop. This effect becomes particularly pronounced at sufficiently low capillary numbers, leading to a strong dependence of the droplet size on the capillary number following a range of low-to-intermediate capillary numbers over which this dependence is significantly weaker; the latter range demarcates the "squeezing" regime, which is attractive from a practical perspective since the drop size only depends on parameters such as the flow rates of the two phases, as well as the channel geometry. This, effectively, provides the motivation for the work: can one provide a scaling theory that allows one to know a priori, with some degree of confidence, the low capillary number limit of the squeezing regime.

I have found this paper to be interesting and competently-written, and organised. I have also found the analysis to be sound (indeed, it is rather simple, once one has understood that the leaking through the gutters is no longer negligible particularly at low capillary numbers). I can see the value of the results presented herein for academics interested in microfluidics (from the practical perspective primarily, though I can also imagine a small proportion of theorists paying attention to this work), though I think the real value is for practitioners. This is also a very well-studied area, and although the results are new, I am left wondering about the impact of this work beyond (the practical end of) microfluidics. For this reason, I think that this work belongs in a more specialised journal such as *Microfluidics Nanofluidics*, *Physical Review Fluids*, etc.

Reviewer #3 (Remarks to the Author):

Review of the manuscript titled "Accounting for corner flow unifies the understanding of droplet formation in microfluidic channels" by Korczyk et al.

The manuscript presented here aims to provide a unified description of droplet production in T-junctions in microchannels. It does so by extending the range of conditions considered (flow rates ratio and capillary numbers) and by considering the leakage flow around the forming droplet. The paper is elegant and the agreement between the data and the models is very good. Also, the authors have addressed many of the theoretical and experimental questions around the problem in a serious way. However there remain a few important points that are not entirely convincing and that the authors should address.

I therefore support publication of this paper in *Nat. Communications* once the authors have addressed the questions below:

1- The authors derive a model based on the minimization of power by the flow in order to obtain Q_N and also the shape parameter S . First of all the manuscript mixes "energy minimization" and "power minimization". Later it also mixes in energy dissipation. This is misleading since power is the derivative of energy and the two terms should not be confused together. The authors should therefore be very precise in what they are minimizing.

More importantly, it is not clear to me that the fluid should really minimize the power at every instant, as they state (p. 3 of the SI). Stokes flows are known to minimize the energy dissipation rate (see e.g. Kim and Karilla's book) but I am not aware of power minimization principles. Therefore the derivation

of Eq. S3, and by the same token Eqs. S12-S15 are a bit suspect since they seem to mix different ideas together. It is critical to clarify what is minimized and to properly justify this in the derivation.

2- One of the most original contributions of this paper is the introduction of fluidic capacitance concepts into microfluidic models (p. 8 around line 80). However it is difficult to get a real physical feeling of what this capacitance is, as the authors stop short of discussing two important aspects: Its relative strength compared to the resistance, and the time-scale that it introduces into the problem. It would be very useful if the authors can give a more quantitative introduction to their C parameter, e.g. by giving its relative strength for different conditions. Also, RC circuits introduce oscillations with a well-defined frequency but it is not clear to me what this would correspond to in the present context.

3- The manuscript makes measurements for different dimensions of square channels but never for different width/height aspect ratios. This is a shame and the authors should at least discuss how the aspect ratio of the channels affects the results discussed here. Ideally they should provide some data for how the device behaves at different aspect ratios.

4- The derivation of the parameter S in the main text is extremely concise. The authors should explain better how the "shape" can be described by a single number, what this number means, and its relation with the actual shape that is observed in the images. Just referring to the SI is too brutal.

More minor points are listed below:

5- The mathematical derivation is difficult to read. The authors should help the reader by introducing paragraph breaks! Additional schematics could also be useful to define all of the different symbols (Q's, K's, V's, ...) that are used here.

6- The reference list mostly ignores droplet microfluidic developments more recent than 2010. It seems important to explicitly state that there have been other approaches to droplet microfluidics than flow-focusing junctions, T-junctions, and co-flow systems that were there from the beginning.

Point by point response

Title: "Accounting for corner flow unifies the understanding of droplet formation in microfluidic channels"

Reviewer #1

This study reports a new regime of droplet formation, leaking regime, which helps enrich the understanding of droplet formation at low Capillary number, the number that compares capillary force with viscous force. The finding of this new regime is made to account for the corner (gutter region of the channels of rectangular) flow and thus its shear impact on droplet formation. In general, the corner flow is neglected when modelling droplet formation in the squeezing regime (low capillary number). The authors found neglecting the corner flow results in failure in predicting droplet formation. This new findings is very impactful to the field and particularly impactful to the applications of droplet microfluidics which normally works with low Capillary numbers.

The study is systematically designed, precisely executed, and accurately and carefully analyzed. For example, when syringe pump is used for pumping the fluids, they carefully looked at its cyclic behavior on droplet formation by varying the syringe size significantly. Another example is that the findings of the leaking regime should disappear when no gutter region is present. The authors actually did perform such an experiment to confirm their findings, which is appreciated. Similar considerations for other aspects were made which is appreciated. The manuscript is well written. I would like to recommend it to be published with the following suggestions to be addressed.

We thank the reviewer for these kind words.

General comments:

1. It's well known that syringe pumps have long-term and short-term oscillations in pumping and one of the authors actually reported the differences between syringe pumps and pressure systems for droplet formation, where pressure systems are much better in producing stable droplet size. It is clear and appreciated that the authors carefully checked the impact of the syringe pump used on droplet formation. The authors should comment on the reason of choosing syringe pumps versus pressure systems.

We are indeed well aware of the limitations of syringe pumps. After confirming that they produce steady flows for the studied range of operating conditions, we decided to use syringe pumps simply because they conveniently allow direct control over the flow rate (rather than converting pressure to flow rate when using pressure pumps). We added this reason to the Methods section.

2. The authors should comment on the reason of choosing much larger channels (800 μm) for images shown in Figure 4 than most channels (360 μm) for other experiments. Was it for better images?

Indeed, the deformations of the shape of the neck are better visible in channels with a rectangular cross section ($W = 2H = 800 \mu\text{m}$) than in channels with a square cross section ($W = H = 360 \mu\text{m}$). We explained this choice and the underlying reason in the caption of Fig. 4b/c. We furthermore added a comparison between rectangular and square channels to the revised ESI.

3. The channel dimensions for most experiments were 360 μm which seems larger than normally used in droplet microfluidics. The rationale of choosing this dimension should be provided.

We added this sentence to the Methods section to explain the rationale:

“These dimensions, which may be larger than those typically encountered, were chosen to enable reaching low Ca values, without being constrained by the lower limit of the flow rate of the syringe pumps.”

4. “very” in the sentence: “Indeed, additional experiments using a T-junction with 'gutter-free' circular channels reveal that the length of the droplets very weakly depends on Ca in comparison to T-junctions with square channels (see ESI, Fig. S.01” should be “vary”.

We changed the phrasing to “varies weakly with”.

Reviewer #2

This paper reports the results of an experimental and scaling study of flows in microfluidic channels with T-junctions. These types of flows exhibit a number of regimes, such as “jetting” and “squeezing” amongst others. This work focuses on a certain regime referred to as “leaking”, on deriving scalings that collapse their experimental data in terms of droplet size/length vs rescaled capillary numbers, and provide transitions between the various regimes. These scaling results are possible by accounting for the fact that drops are not completely flush against the square channel walls; there are “gutter”-like regions surrounding the droplet through which a proportion of the continuous phase flow can by-pass the droplet. This, therefore, implies that not all of the continuous phase liquid can contribute to the squeezing that gives rise to the pinch-off event, which, in turn, leads to the formation of a discrete drop. This effect becomes particularly pronounced at sufficiently low capillary numbers, leading to a strong dependence of the droplet size on the capillary number following a range of low-to-intermediate capillary numbers over which this dependence is significantly weaker; the latter range demarcates the “squeezing” regime, which is attractive from a practical perspective since the drop size only depends on parameters such as the flow rates of the two phases, as well as the channel geometry. This, effectively, provides the motivation for the work: can one provide a scaling theory that allows one to know a priori, with some degree of confidence, the low capillary number limit of the squeezing regime.

I have found this paper to be interesting and competently-written, and organised. I have also found the analysis to be sound (indeed, it is rather simple, once one has understood that the leaking through the gutters is no longer negligible particularly at low capillary numbers). I can see the value of the results presented herein for academics interested in microfluidics (from the practical perspective primarily, though I can also imagine a small proportion of theorists paying attention to this work), though I think the real value is for practitioners. This is also a very well-studied area, and although the results are new, I am left wondering about the impact of this work beyond (the practical end of) microfluidics. For this reason, I think that this work belongs in a more specialised journal such as *Microfluidics Nanofluidics*, *Physical Review Fluids*, etc.

We thank the reviewer for these kind words. We agree that this work will certainly be of immediate impact for practitioners. On top of that, our work contains an elegant lesson in physics. With the analysis based on simple frameworks (e.g. electric circuit, variational principle), we expect that the educational example of a simple model that explains a complex physical process will be appealing to an audience much broader than the microfluidics and soft matter communities.

Our work is directly connected to the classical example of shear emulsification thought in fluid mechanics and illustrates how confinement alters the interplay between viscous and interfacial forces, in an unexpected way. We therefore believe that the proposed publication will significantly widen the understanding of two-phase flows in geometrical confinement, which are encountered in numerous applications.

Reviewer #3

The manuscript presented here aims to provide a unified description of droplet production in T-junctions in microchannels. It does so by extending the range of conditions considered (flow rates ratio and capillary numbers) and by considering the leakage flow around the forming droplet. The paper is elegant and the agreement between the data and the models is very good. Also, the authors have addressed many of the theoretical and experimental questions around the problem in a serious way. However there remain a few important points that are not entirely convincing and that the authors should address. I therefore support publication of this paper in Nat. Communications once the authors have addressed the questions below:

We thank the reviewer for these positive words and for the sound and constructive remarks provided below.

1- The authors derive a model based on the minimization of power by the flow in order to obtain Q_N and also the shape parameter S . First of all the manuscript mixes "energy minimization" and "power minimization". Later it also mixes in energy dissipation. This is misleading since power is the derivative of energy and the two terms should not be confused together. The authors should therefore be very precise in what they are minimizing.

We fully agree that the previous derivation of the shape parameter S was confusing. We thoroughly revised the derivation by basing it on the variational principle as proposed by Lars Onsager. This principle is an extension of Rayleigh's principle of least energy dissipation and has been successfully applied to the analysis of the time evolution of non-equilibrium systems, particularly in problems of soft matter physics (some of the examples can be found in the references).

Please notice that the final form of the thus obtained parameter S is the same as in the original manuscript.

More importantly, it is not clear to me that the fluid should really minimize the power at every instant, as they state (p. 3 of the SI). Stokes flows are known to minimize the energy dissipation rate (see e.g. Kim and Karilla's book) but I am not aware of power minimization principles. Therefore the derivation of Eq. S3, and by the same token Eqs. S12-S15 are a bit suspect since they seem to mix different ideas together. It is critical to clarify what is minimized and to properly justify this in the derivation.

We fully agree that minimization of the energy dissipation rate is the proper framework to set up the analysis. We thoroughly revised the S -number derivation, as mentioned above. We removed Eq. S3, and Eqs. S12-S15 from the ESI, because we included the complete derivation of S -number in the main text.

2- One of the most original contributions of this paper is the introduction of fluidic capacitance concepts into microfluidic models (p. 8 around line 80). However it is difficult to get a real physical feeling of what this capacitance is, as the authors stop short of discussing two important aspects: Its relative strength compared to the resistance, and the time-scale that it introduces into the problem. It would be very useful if the authors can give a more quantitative introduction to their C parameter, e.g. by giving its relative strength for different conditions. Also, RC circuits introduce oscillations with a well-defined frequency but it is not clear to me what this would correspond to in the present context.

We thank the reviewer for acknowledging this original contribution. In the manuscript, we provide the following physical interpretation of the fluidic capacitance: *“Adopting the electric-hydraulic analogy that can be applied to single phase low Reynolds number flows, we describe the accumulation of the CP behind the droplet akin to a charging capacitor: the further the rear interface is pushed into the junction, the larger the difference in curvatures at the back and front, and hence the more charge is stored. If one were able to instantaneously release the driving pressures in the system during the formation of a droplet (akin to switching off the main voltage supply in an electric circuit), the capacitor would discharge, i.e., the forming droplet would relax its shape inside the T-junction to an equilibrium shape with similar curvatures at its front and back.”*

We completed this comparison by adding the following sentence on the introduction of the time scale: *“Similar to electric RC circuits, the product of the capacitance and resistance can be seen as the characteristic relaxation time. A quantitative analysis of this time scale is provided in the ESI.”*

We furthermore added an intuitive physical interpretation of the fluidic capacitance C using the following sentences: *“Before using the thus obtained relation, we provide an intuitive physical interpretation of the capacitance by connecting it to the classical derivation of the Young-Laplace law. Considering a droplet of volume V and surface area A, the Helmholtz free energy, dF, equals $dF = -pdV + \gamma dA$. Near equilibrium ($dF = 0$), the pressure difference across the droplet interface equals $\Delta p = \gamma \frac{dA}{dV} = \gamma K$. The curvature difference for a forming droplet, $K_F - K_B$, being equal to $\frac{V_N^*}{c}$ can similarly be related to $\frac{dA_N(V_N^*)}{dV_N^*}$, i.e. the change in neck area, A_N , parameterized solely in terms of V_N^* . We hence obtain $\frac{dA_N(V_N^*)}{dV_N^*} \sim \frac{V_N^*}{c}$, with the corresponding interfacial free energy, \mathcal{W} , being equal to $\mathcal{W} = \gamma A_N(V_N^*) = \frac{1}{2} \frac{(V_N^*)^2}{c}$. The capacitance C can hence be interpreted as the coefficient that describes how the surface area of the neck, A_N , and the interfacial free energy stored in it due to its deformation by the CP, depends on the volume of the CP collected behind the neck. Now, we proceed with the main analysis by substituting the relation $K_F - K_B = \frac{1}{W} \frac{v_N^*}{c}$ [...]”.*

On top of the addition of the conceptual interpretation of C, we added a quantitative description of the time scales governing the problem to the ESI, including the time scale introduced by the resistance and capacitance, based

on an analysis of Eq. S1. This quantitative analysis teaches that this time scale is negligible compared to the time scale introduced by the conditions for the leaking regime, while being of the same order in the squeezing regime. Importantly, for the conditions studied in this work, the time scales only lead to a short transient, after which Q_n reaches a constant value. The resulting droplet size hence does not depend on the time scale introduced by the capacitance. This is now clarified in the ESI, below Eq. S1.

Regarding oscillations: while oscillations may occur in RLC system, they are not expected in RC systems.

3- The manuscript makes measurements for different dimensions of square channels but never for different width/height aspect ratios. This is a shame and the authors should at least discuss how the aspect ratio of the channels affects the results discussed here. Ideally they should provide some data for how the device behaves at different aspect ratios.

We fully agree that a discussion on the effect of aspect ratio is of interest. Following the reviewer's suggestion, we added a comparison of data obtained in channels with different aspect ratios ($H/W = 0.5$ and $H/W = 1$) to the ESI (Supplementary Figure 5 and Supplementary Table 4). We furthermore added the following sentence to the main text: *"For rectangular channels with different aspect ratios, we confirm that the general behaviour is the same, with the details of the leaking mechanisms depending on the aspect ratio (Supplementary Figure 5 and Supplementary Table 4)."*

An extensive study on the effect of cross-sectional geometry, including channel aspect ratio for rectangular channels, is currently performed in our lab and beyond the scope of the present work.

4- The derivation of the parameter S in the main text is extremely concise. The authors should explain better how the "shape" can be described by a single number, what this number means, and its relation with the actual shape that is observed in the images. Just referring to the SI is too brutal.

We fully agree. We removed the derivation of the S -number from the ESI and incorporated an expanded and thoroughly revised derivation in the main article.

More minor points are listed below:

5- The mathematical derivation is difficult to read. The authors should help the reader by introducing paragraph breaks! Additional schematics could also be useful to define all of the different symbols (Q 's, K 's, V 's, ...) that are used here.

We wrote the derivation from scratch, basing it on Onsager's variation principle. We incorporated the full derivation in the main article, aiming for a text that is easy to read and understand.

6- The reference list mostly ignores droplet microfluidic developments more recent than 2010. It seems important to explicitly state that there have been other approaches

to droplet microfluidics than flow-focusing junctions, T-junctions, and co-flow systems that were there from the beginning.

We agree with the importance to acknowledging other (recent) approaches and added the following references:

4. Baroud, C. N., Gallaire, F. & Dangla, R. Dynamics of microfluidic droplets. *Lab. Chip* (2010). doi:10.1039/c001191f
18. Nekouei, M. & Vanapalli, S. A. Volume-of-fluid simulations in microfluidic T-junction devices: Influence of viscosity ratio on droplet size. *Phys. Fluids* **29**, 032007 (2017)
19. Chakraborty, I., Ricouvier, J., Yazhgur, P., Tabeling, P. & Leshansky, A. M. Droplet generation at Hele-Shaw microfluidic T-junction. *Phys. Fluids* **31**, 022010 (2019).
22. Chen, X., Glawdel, T., Cui, N. & Ren, C. L. Model of droplet generation in flow focusing generators operating in the squeezing regime. *Microfluid. Nanofluidics* **18**, 1341–1353 (2014).
23. Umbanhowar, P. B., Prasad, V. & Weitz, D. A. Monodisperse Emulsion Generation via Drop Break Off in a Coflowing Stream. *Langmuir* **16**, 347–351 (2000).
24. Kawakatsu, T., Kikuchi, Y. & Nakajima, M. Regular-sized cell creation in microchannel emulsification by visual microprocessing method. *J. Am. Oil Chem. Soc.* **74**, 317–321 (1997).
25. Dangla, R., Fradet, E., Lopez, Y. & Baroud, C. N. The physical mechanisms of step emulsification. *J. Phys. Appl. Phys.* **46**, 114003 (2013).
26. Dangla, R., Kayi, S. C. & Baroud, C. N. Droplet microfluidics driven by gradients of confinement. *Proc. Natl. Acad. Sci. U. S. A.* **110**, 853–858 (2013).
27. Li, Z., Leshansky, A. M., Pismen, L. M. & Tabeling, P. Step-emulsification in a microfluidic device. *Lab. Chip* **15**, 1023–1031 (2015).
28. Chakraborty, I., Ricouvier, J., Yazhgur, P., Tabeling, P. & Leshansky, A. M. Microfluidic step-emulsification in axisymmetric geometry. *Lab. Chip* **17**, 3609–3620 (2017).
29. Eggersdorfer, M. L., Seybold, H., Ofner, A., Weitz, D. A. & Studart, A. R. Wetting controls of droplet formation in step emulsification. *Proc. Natl. Acad. Sci.* **115**, 9479–9484 (2018).
30. Nisisako, T. & Torii, T. Microfluidic large-scale integration on a chip for mass production of monodisperse droplets and particles. *Lab. Chip* **8**, 287–293 (2008).
31. Amstad, E. et al. Robust scalable high throughput production of monodisperse drops. *Lab. Chip* **16**, 4163–4172 (2016).

32. Amstad, E. et al. Parallelization of microfluidic flow-focusing devices. *Phys. Rev. E* 95, 043105 (2017).
35. Kaminski, T. S., Scheler, O. & Garstecki, P. Droplet microfluidics for microbiology: techniques, applications and challenges. *Lab. Chip* 16, 2168–2187 (2016).
36. Guo, M. T., Rotem, A., Heyman, J. A. & Weitz, D. A. Droplet microfluidics for high-throughput biological assays. *Lab Chip* 12, 2146–2155 (2012).
37. Klein, A. M. et al. Droplet Barcoding for Single-Cell Transcriptomics Applied to Embryonic Stem Cells. *Cell* 161, 1187–1201 (2015).
41. Göke, K. et al. Novel strategies for the formulation and processing of poorly water-soluble drugs. *Eur. J. Pharm. Biopharm.* 126, 40–56 (2018).
56. Doi, M. Onsager's variational principle in soft matter. *J. Phys. Condens. Matter* 23, 284118 (2011).
57. Zhou, J. & Doi, M. Dynamics of viscoelastic filaments based on Onsager principle. *Phys. Rev. Fluids* 3, 084004 (2018).
58. Oh, K. W., Lee, K., Ahn, B. & Furlani, E. P. Design of pressure-driven microfluidic networks using electric circuit analogy. *Lab. Chip* 12, 515–545 (2012).

REVIEWERS' COMMENTS:

Reviewer #1 (Remarks to the Author):

The revised manuscript has addressed my previous concerns. The authors' responses to the other reviewers' comments are also appropriate. I would like to recommend it to be published.

Reviewer #2 (Remarks to the Author):

The authors have convinced me that the paper is worth publishing in Nat. Comms. I am happy to recommend it for publication.

Point by point response to issues raised by referees

Title: "Accounting for corner flow unifies the understanding of droplet formation in microfluidic channels"

MS#: NCOMMS-18-35904A

REVIEWERS' COMMENTS:

Reviewer #1 (Remarks to the Author):

The revised manuscript has addressed my previous concerns. The authors' responses to the other reviewers' comments are also appropriate. I would like to recommend it to be published.

Reviewer #2 (Remarks to the Author):

The authors have convinced me that the paper is worth publishing in Nat. Comms. I am happy to recommend it for publication.

We are happy that our final revision of the article meets all the requirements of reviewers.

We thank the reviewers for kind words and for finding our work suitable for the publication in Nature Communications.